# Effects of Concentration of Soybean Protein Isolate and Maltose and Oil Phase Volume Fraction on Freeze–Thaw Stability of Pickering Emulsion

**DOI:** 10.3390/foods11244018

**Published:** 2022-12-12

**Authors:** Ziyue Song, Yang Yang, Fenglian Chen, Jing Fan, Bing Wang, Xin Bian, Yue Xu, Baoxiang Liu, Yao Fu, Yanguo Shi, Xiumin Zhang, Na Zhang

**Affiliations:** College of Food Engineering, Harbin University of Commerce, Harbin 150076, China

**Keywords:** pickering emulsion, freeze–thaw stability, nanoparticles concentration, dispersed phase volume fraction, stability analysis

## Abstract

There is growing interest in enhancing the freeze–thaw stability of a Pickering emulsion to obtain a better taste in the frozen food field. A Pickering emulsion was prepared using a two-step homogenization method with soybean protein and maltose as raw materials. The outcomes showed that the freeze–thaw stability of the Pickering emulsion increased when prepared with an increase in soybean protein isolate (SPI) and maltose concentration. After three freeze–thaw treatments at 35 mg/mL, the Turbiscan Stability Index (TSI) value of the emulsion was the lowest. At this concentration, the surface hydrophobicity (H_0_) of the composite particles was 33.6 and the interfacial tension was 44.34 mN/m. Furthermore, the rheological nature of the emulsions proved that the apparent viscosity and viscoelasticity of Pickering emulsions grew with a growing oil phase volume fraction and concentration. The maximum value was reached in the case of the oil phase volume fraction of 50% at a concentration of 35 mg/mL, the apparent viscosity was 18 Pa·s, the storage modulus of the emulsion was 575 Pa, and the loss modulus was 152 Pa. This research is significant for the production of freeze–thaw resistant products, and improvement of protein-stabilized emulsion products with high freeze–thaw stability.

## 1. Introduction

Frozen fast food has received wide attention due to consideration of the needs inherent to a fast-paced life and the demand for better quality of products [1]. Freezing is a broadly used method that improves the shelf life of provisions, while the ice crystals composed throughout the freezing process constitute an important environmental stress, causing damage to the structure of provisions and ultimately generating an influence on the quality of products [2,3]. Therefore, it is worth producing emulsion-based products with an enhanced freeze–thaw stability. Pickering emulsion has excellent biocompatibility and a strong anti-aggregation ability. Solid particles adsorbed at the interface between oil and water in a Pickering emulsion can form a thick viscoelastic layer around the surface of oil droplets. During the freezing process, the viscoelastic layer in the Pickering emulsion resist the stress caused by ice expansion, giving it a certain freeze–thaw stability, and thus guaranteeing good product quality after the freezing treatment [4]. At present, the methods used to prepare a Pickering emulsion with high freeze–thaw stability include heating modification of protein, modification of the protein glycosylation, adjustment of ionic strength, and so on. For example, a steady thymol oil-in-water Pickering emulsion was prepared through electrostatic interaction between soluble almond gum and whey protein isolation. The combination of whey protein and soy polysaccharide using heat-induced polymerization was used to strengthen the freeze–thaw stability of the emulsion [5]. The use of three various vegetable oils in the freeze–thaw stability of Pickering emulsion prepared with quinoa protein, determined that various vegetable oils exerted little effect on the freeze–thaw stability of Pickering emulsion [6].

Bean protein can be used as a substitute for animal protein due to its low cost and excellent performance. Common bean proteins are soybean protein and pea protein. Soybean protein has good emulsifying properties and solubility. In addition, the amphiphilic nature of soybean protein makes it an excellent interface stabilizer. Therefore, this experiment takes soybean protein as the experimental material [7,8]. However, a Pickering emulsion prepared with soy protein in its natural state has poor freeze–thaw stability and is difficult to be widely used in frozen foods [9]. This is because when the external temperature decreases, water forms ice crystals, and the stress generated by the ice crystals plays a destructive role in the solid particles adsorbed at the oil–water interface; thus, the stability of the emulsion is destroyed [10]. Maltose has good frost resistance and thermal stability, ensuring that the Maillard reaction is in the early stages. By adding maltose, the carbonyl group of the sugar and the amino group of the soy protein undergo a condensation reaction [11]. The electrostatic repulsion increased with the charge density of the soy protein changing. Changing the amphiphilicity of the soy protein causes it to have balanced wettability and strengthening. During the effective adsorption of sugar particles at the oil–water interface, the freeze–thaw stability of the emulsion is improved [12,13,14].

The thermally induced aggregation method is a simple and effective method for preparing proteins and sugars to form nanoparticles, nanofibrils and nanogels. It has the advantages of a simple preparation and operation process, low cost, avoiding chemical residues, etc [10,15]. Therefore, this experiment adopts the heat-induced aggregation method to combine soybean protein and maltose to enhance its freeze–thaw stability. At present, the effect of soy protein, maltose concentration, and oil and water ratio of a Pickering emulsion on freeze–thaw stability has still not been studied. This study aims to prepare a a Pickering emulsion with high freeze–thaw stability, which can support the subsequent improvement in frozen food quality.

## 2. Materials and Methods

### 2.1. Materials

SPI was obtained from Yuwang Industrial Co., Ltd. (Dezhou, China). Maltose (purity ≥ 95%) was obtained from Ding sheng Industrial Co., Ltd. (Guangzhou, Guangzhou Province, China). Soy oil was accessed through a supermarket in Harbin, China. Other reagents were analyzed and bought from Shangdong Tuopu Biotechnology Co., Ltd. (Qingdao, China).

### 2.2. Preparation of Dispersions Containing Soy Protein Isolate and Maltose

SPI–maltose particles were created with reference to the approaches of Cabezas et al. with slight modifications [16]. The solutions with SPI and maltose concentrations of 15, 20, 25, 30, 35 mg/mL were made, and then mixed at equal concentration and volume at a proportion of 1:1, and magnetically stirred for 2 h after mixing. The SPI–maltose solution and the maltose solution were heated at 90 °C for 15 min and were rinsed and cooled immediately to obtain the SPI–maltose complexes. Afterwards, they were stored overnight at 4 °C (for the full hydration of SPI–maltose). The pH of the solution was changed to 8. The 0.02% NaN_3_ (*w*/*v*) was put into control for the development of microorganisms in the emulsions.

### 2.3. Characterization of SPI–maltose Particles

The methods for measuring Dz and potential referred to Chen et al.’s, with slight modifications [17]. The test sample was diluted with deionized water to an SPI–maltose solution of about 0.01 mg/mL, and was then screened through a 0.45 μm membrane filter. The refractive index of SPI–maltose is 1.460, and was tested using a Zetasizer Nano S90 Malvern particle size analyzer (Malvern Instruments Ltd., Worcester, UK).

### 2.4. Determination of SPI–maltose H_0_

The method for measuring the hydrophobicity (H_0_) of the particle surface was determined pursuant to the approach of Xu et al., using an F97pro fluorescence spectrophotometer (Shanghai Lengguang Technology Co., Ltd., Shanghai, China) [18]. The SPI–maltose samples (0.01 mol/L) were dissolved in pH 7.0 phosphate buffer, which was followed by 1 hour stirring at room temperature and 30-min centrifugation at 10,000× *g* centrifugal force. To the sample solutions of 4 mL were added 8 mmol/L ANS solution of 40 L, and these were shaken and set for 3 min.

### 2.5. Interfacial Behavior of Nanoparticles

The interfacial tension was measured as described by Feng et al., with slight modifications [19]. Surface tension (MC-1021 homogenizer, Mince Instrument Equipment Co., Ltd., Xiamen, China) meter assayed the interfacial tension of the sample. The samples were poured into the measuring cup to the middle graduation line. Using the platinum plate method, the interfacial tension of each sample was measured 3 times with an automatic interfacial tension meter at 25 °C.

### 2.6. Preparation of Pickering Emulsions

The preparation method of a Pickering emulsion was used with reference to Zhu et al. [20]. The soybean oil (10%, 20%, 30%, 40%, 50% *w*/*w*) was added to SPI–maltose complexes with a concentration of 35 mg/mL. The mixed solutions were stirred for 3 min at a speed of 12,000 rpm through an XHF-DY high-speed homogenizer (Xinzhi Biological Co., Ltd., Ningbo, China). The crude emulsion was further homogenized through a Scientz-150 high-pressure homogenizer (Xinzhi Biological Co., Ltd., Ningbo, China) at a stress degree of 80 MPa to produce an initial emulsion. The Pickering emulsion (40 g) was placed in a 100 mL plastic bottle, which was followed by 22 h freezing at −22°C and 2 h thawing at 30 °C. The freeze–thaw cycle treatment was repeated 3 times to obtain a targeted emulsion.

### 2.7. Stability Analysis of Pickering Emulsion

As stated in the research of Gmach et al. [21], a Turbiscan stability analyzer (Formulaction, Toulouse, France) was adopted to detect the stability of the initial emulsion and the freeze–thaw process. The emulsion of 20 mL was put in a bottle and scanned from the bottom to top at room temperature (25 °C) for 24 h. The diffuse light changes of the initial emulsion and freeze–thaw treatment emulsion were determined.

### 2.8. Confocal Laser Scanning Microscope (CLSM)

CLSM (Leica Microsystems Inc., Heidelberg, Germany) was adopted to review varieties of the microstructure of the Pickering emulsion in the whole freeze–thaw cycle with the test approach of Zhu et al. [22]. Briefly, the protein stage of emulsion was dyed with Nile Blue, while the oil stage of it was dyed with Nile Red for 30 min. Stained Pickering emulsions were dripped onto glass slides and monitored by CLSM.

### 2.9. Rheological Measurement

The rheometer (MCR102, Antona Co., Ltd., Graz, Austrian) was selected to test the shear viscosity of the different emulsions and select the viscoelastic fluid mode. The shear rate is from 0.05 to 100 s^−1^.

The viscoelasticity of different emulsions was measured by the rheological strain scan and frequency scan mode using a rheometer (MCR102, Antona Co., Ltd., Graz, Austrian). The strain scan range is from 0.001–1.0, and the frequency is set to 1 Hz. The measurement results expressed in the storage modulus (G′) and the loss modulus (G″) were recorded.

### 2.10. Differential Scanning Calorimetry (DSC)

The 204 F1 DSC (NETZSCH-Gerätebau GmbH, Selb, Germany) measurement method was modified as stated by the previous study [23]. A small amount of the Pickering emulsion (10–20 mg) was placed on a dish. The temperature set range was from −50 °C to 40 °C. The cool temperature set range was from 40 °C to −50 °C at the rate of 5 °C/min, balanced and then heated to 40°C at the same rate.

### 2.11. Statistical Analysis

The above experiments were carried out at least in triplicate, and the origin 2019 software was adopted for exploration and mapping. SPSS 20 software was adopted for one-way ANOVA and Duncan’s test of the experimental data, with *p* < 0.05 as the great difference, and the outcomes were expressed by mean ± SD.

## 3. Results

### 3.1. Particle Size and Potential Change of SPI–maltose

By measuring the complex potential and particle size of SPI–maltose at different concentrations, the properties of SPI–maltose particles were evaluated. Figure 1A shows the particle size (Dz) of SPI–maltose nanoparticles when they were developed at the concentration values of 15–35 mg/mL. The results proved the Dz of the produced composite particle gradually increased, which conformed to the findings of Chen et al.’s research [24]. Within the concentrations of 15–35 mg/mL, particle diameter increased with the increase in the concentrations, and a larger Dz was beneficial to obtain higher freeze–thaw stability for the Pickering emulsion. It was observed that the particle concentration had little impact on the potential, and the potential of the composite particles under different SPI–maltose concentrations was in the range of −27 to −29 mV (Figure 1B).

### 3.2. The H_0_ of SPI–maltose

The interaction between two water-insoluble molecules was the key indicator to gauge the working feature of proteins. The H_0_ of composite particles gradually increased with the increase in SPI and maltose concentrations (Figure 2). When the concentration reached 35 mg/mL, the H_0_ reached the maximum. Soy protein’s conformation was changed by the interaction between SPI and maltose. Glycosylation causes reduced H_0_ of the SPI, and heating during the Maillard reaction causes the unfolding of SPI conformation, thereby exposing the internal polar groups and reducing the hydrophobicity on the SPI surface [25]. However, as the concentration of the SPI increases, the hydrophobic amino acid content in the complex increases, which in turn increases the H_0_ of the complex particles [26].

### 3.3. The Interfacial Tension of SPI–maltose

Interfacial tension was the prerequisite for emulsification and foam generation, and a reduced level was closely relevant to the emulsifying and foaming capabilities. The interfacial tension of SPI–maltose nanoparticles was measured with an interfacial tension meter, and the outcomes were proven in Figure 3. The interfacial tension decreased from 46.62 mN/m to 44.34 mN/m, while the particle concentration rose from 15 mg/mL to 35 mg/mL. The results showed that increasing the concentration of SPI and maltose could reduce the interfacial tension. This was because, with the increase in concentration, the concentration of SPI–maltose molecules increased as well, which increased the possibility of aggregation of SPI–maltose. The enthalpy of aggregation may be lower than that of a single molecule, which made it tend to be adsorbed at the interface, thus reducing the interfacial tension. In addition, the contact probability between SPI-free amino groups and carbonyl groups at the reducing end of maltose increased, which promoted the reaction between SPI and maltose, leading to more SPI expansion and higher SPI flexibility, as well as reducing the interfacial tension of SPI–maltose molecules in oil and water. In addition, the contact probability between the free amino group and the carbonyl group of SPI at the reducing end of maltose increased, promoting the reaction, leading to more SPI unfolding and higher SPI flexibility, and reducing the interfacial tension of SPI–maltose molecules in oil and water. Lower interfacial tension can promote the formation rate of bridges between droplets and shorten the contact time between droplets. The shorter the contact time between droplets, the better the emulsion’s stability, as it can better resist the pressure brought by ice crystals and improve the freeze–thaw stability of the emulsion [27].

### 3.4. Pickering Emulsion’s Freeze–thaw Stability

Turbiscan stability analyzer is one of the methods that can quickly and reliably measure the stability of the Pickering emulsion. The freeze–thaw stability of the emulsion was evaluated by the change of TSI and backscattered light intensity (ΔBS) (Figure 4).

Firstly, the slope of the TSI curve of the Pickering emulsion after freeze–thaw cycle was higher than that of the initial emulsion, indicating that freezing would damage the Pickering emulsion and reduce its stability. This is because when the Pickering emulsion was frozen, oil phase or water phase in the Pickering emulsion crystallized to make the droplets gather, which destroyed the film at the oil–water interface of the Pickering emulsion, making the Pickering emulsion easier to agglomerate during the dissolution process, and thus reducing its stability [21]. By increasing the concentration of SPI and maltose, the overall TSI of the initial emulsion was reduced from 4.9 to 1.8. The TSI of the prepared Pickering emulsion after three freeze–thaw cycles was 2.9 when the oil volume was 40% and the concentration was 35 mg/mL, which still showed good stability. The reason for this phenomenon was that, at low particle concentration, the emulsion was unstable because the droplet was partially covered by SPI–maltose nanoparticles, and there was almost no excess SPI–maltose particles in the continuous phase. By increasing SPI–maltose concentration, the aggregate of excessive SPI–maltose particles in the continuous phase changed from dense flocs to loose networks, so that the droplets kept a good separation from each other. On the other hand, the lower SPI and maltose concentration might form a thin polymer layer on the surface of the droplets, resulting in the easy aggregation and coalescence of the droplets. As oil phase fraction increased, the overall TSI of the initial emulsion reduced from 4.7 to 0.2. When the concentration was 35 mg/mL and oil phase was 50%, the TSI of freeze–thaw treatment for three times was 0.7. At this time, the Pickering emulsion still had a better stability. This was because when dispersed phase volume fraction increased within a certain range, more oil droplets in the emulsion could be filled into the emulsion layer, and the crosslinked gel structure was tighter, which made the Pickering emulsion system more stable so that it could resist the damage caused by ice crystals during freezing [28].

The above experimental results can be further confirmed by ΔBS (Figure 5 and Figure 6). ΔBS can evaluate the change of concentration or droplet size in the Pickering emulsion. In Figure 5, it is proved that the Pickering emulsion ΔBS changes with different concentrations of SPI and maltose before and after a freeze–thaw cycle treatment. After freeze–thaw cycle treatment of emulsion, with the increase in freeze–thaw cycle times, ΔBS increased gradually; however, as the concentration increased, ΔBS gradually dropped from −65% to −20% at the bottom of freeze–thaw lotion, while ΔBS decreased from 36% to 10%. The results show that higher particle concentration (35 mg/mL) can improve its ability to stabilize emulsion, and the emulsion droplets are not easy to flocculate or coalesce. The emulsion formed by higher particle concentration still has a certain stability after freeze–thaw cycle treatment. It may be that, with growing concentration, the H_0_ of particles increased and the interfacial tension decreased, which is conducive to more effective adsorption of particles at the oil–water interface, so as to improve freeze–thaw stability of emulsions [29,30].

To illustrate the potential mechanism of improving the stability of the emulsion by increasing dispersed phase volume fraction, Figure 6 shows the stability of the initial emulsion and the freeze–thaw emulsion with different dispersed phase volume fraction ΔBS. With the dispersed phase volume fraction doubling in the range of 10–50%, the ΔBS gradually decreased from −65% to −20%, while the top ΔBS decreased from about 35% to 10% (Figure 6). This phenomenon, combined with the rheological results of the emulsion, shows that the interaction between oil droplets affects the viscoelasticity of the emulsion. By increasing the stiffness of the emulsion’s gel network structure, the freeze–thaw stability of the emulsion can be improved. Therefore, the volume fraction of oil phase and particle concentration can be appropriately increased to make the emulsion have better freeze–thaw stability [31].

### 3.5. The CLSM of Pickering Emulsion

According to the results of stability analysis, the initial emulsion and frozen processing emulsion under the best conditions of the Pickering emulsion frozen stability were selected. The oil phase indicator was red, and the protein indicator was green. The CLSM (Figure 7) showed that the Pickering emulsion formed under this condition still guaranteed its own stability after one and two freeze–thaw cycles. As stated by the outcomes of protein staining, the protein’s fluorescence intensity was weak and a large amount of aggregation was not found, indicating that the protein was in a state of adsorption at the oil–water interface and the stability of emulsion was excellent. The oil phase showed obvious large particles aggregated, and the protein signal was meaningfully enhanced and appeared aggregated when the samples carried out three freeze–thaw cycles. The particle layer formed by irreversibly adsorbed particles on the interface physically blocks the coalescence of droplets and the sharing of gel particles by oil droplets. It can spatially obstruct the agglomeration of droplets through bridging and ensure the stability of the emulsion; thereby, the freeze–thaw stability was enhanced. That is, the freeze–thaw stability of Pickering emulsion may be consequently enhanced by adapting particles’ adsorption at the oil–water interface [23].

### 3.6. Rheological Properties of the Pickering Emulsion

In order to clearly show the mechanism of freeze–thaw stability of the Pickering emulsion, the apparent viscosity, G′ and G″, of the initial Pickering emulsions was characterized. The apparent viscosity of the emulsion prepared with SPI–maltose at different concentrations and different oil volume fractions was shown in Figure 8. As shear rate increased, the fluidity of the emulsion declined, and the viscosity of the initial emulsion grew with the increased concentration. When the concentration was 35 mg/mL, the apparent viscosity of the Pickering emulsion increased as the volume proportion of oil phase increased. When the particle concentration was 35 mg/mL and the volume of oil phase was 50%, the maximum apparent viscosity was 18 Pa·s. This is because the increase in SPI–maltose concentration and oil volume fraction enhanced the adhesion between the Pickering emulsion droplets, and the spatial network structure between the emulsion is more compact, causing it to show greater apparent viscosity [32,33].

Secondly, the apparent viscosity of the emulsion at different concentrations of SPI– maltose and dispersed phase volume fractions under different shear rate γ were determined, and fitted the apparent viscosity of the emulsion according to the power law model. R_2_ in the emulsion fitting equation is 0.98–0.995, indicating that the flow curve at this time is more in line with the power law model. The consistency parameter K and flow index in the power law equation were empirical constants related to the properties of liquid. K is a measure of liquid viscosity; the larger the K, the more viscous the emulsion. n value was a measure of the degree of pseudoplasticity. If n value is less than 1, it is a shear thinning pseudoplastic fluid [34]. The greater the deviation of n value from Table 1, the easier the shear thinning is; that is, the greater the degree of pseudoplasticity. It was found from Table 1 and Table 2 that the flow index of all samples is *n* < 1, showing pseudoplastic fluid properties. As the shear rate increased, the apparent viscosity of the Pickering emulsion composed of various components dropped sharply, leading to the shear thinning effect. Firstly, the solid particles in the emulsion were adsorbed on the oil–water interface, and the oil droplets were dispersed in the water. However, the components flow at different speeds, and the emulsion could not keep this state for long. When the emulsion was flowing, every long chain molecule would enter the flow layer with the same flow rate as far as possible. Secondly, the SPI–maltose molecule had a long molecular chain. In the static state, these substances maintained irregular internal order and had high internal resistance to hinder their movement. With the increase in shear rate, the rearrangement of substances led to the decrease in resistance and apparent viscosity of emulsions. With the increase in SPI and maltose concentration, the volume fraction of the oil phase increased, the irregular internal order was maintained, the viscosity increased, and the internal resistance was large, which hindered the flow of emulsion. However, as the SPI–maltose concentration increased and the volume fraction of the oil phase grew, the internal irregular order is maintained, the viscosity increases, and the internal resistance is large, thus hindering the flow of the emulsion. It shows that increasing the concentration of SPI and maltose and increasing the volume ratio of the oil phase are beneficial to enhancing the fluid nature of the Pickering emulsion and improving the freeze–thaw stability of Pickering emulsion [35,36].

In addition, the G′ of all emulsion is greater than the G″, indicating that a gel network structure guided by spring is formed in the emulsion system. At a frequency in the scope of 0.1–100 rad/s, the G′ and G″ of initial emulsions grew in line with the increase in particle concentrations and the dispersed phase volume fraction (Figure 9). When the concentration was 35 mg/mL and the volume fraction of oil phase was 50%, the maximum value of G″ was 100 Pa and the maximum value of G′ was 575 Pa. The outcomes showed that the viscoelasticity of the Pickering emulsion could be improved, the damage of ice crystals to the emulsion could be reduced, and the freeze–thaw stability of the Pickering emulsion could be improved by increasing the concentration and dispersed phase volume fraction to a certain extent [37]. The crosslinking of protein molecules increases the penetration resistance of the interface layer during thawing and freezing, thus efficiently controlling the migration of oil droplets and strengthening the freeze–thaw stability of the emulsion gel [2].

### 3.7. Pickering Emulsion’s Thermal Properties

The alteration of the crystallization point for the emulsion was also one of the methods used to improve the Pickering emulsion’s freeze–thaw stability [38]. To understand the effects of SPI and maltose concentration and oil–water ratio on the temperature of crystallization and melting during freezing of SPI–M preparation of the Pickering emulsion, we recorded the DSC thermal spectrum of the Pickering emulsion. Figure 10A shows the DSC thermal spectrum of the Pickering emulsion prepared at different concentrations of SPI and maltose, and Figure 10B is the DSC thermal spectrum of the Pickering emulsion prepared with different oil phase volume fractions.

Figure 10A shows that the crystallization temperature of the Pickering emulsion prepared under different concentrations of SPI and maltose was −12.05 to −14.29 °C. As the concentration grew, the crystallization temperature of the emulsion first declined and then increased. This may be because the increase in SPI and maltose concentration accelerated the reaction speed, increased the particle size of SPI–maltose particles, increased the number of nucleation in the reaction, and lowered the crystallization temperature of the emulsion. In addition, the high concentration of SPI and maltose formed a thick interface film at the interface, which provided great resistance to the phase transition, thus improving the freeze–thaw stability of the emulsion. In addition, the high concentration of SPI and maltose formed a thick interface film at the interface, which provided great resistance to the phase transition, thus improving the freeze–thaw stability of the emulsion. According to Figure 10B, the crystallization temperature of the Pickering emulsion prepared by different oil ratios is from −11.17 to −15.22 °C, which may be due to the crystallization temperature of soybean oil being −15 °C; thus, proper addition of soybean oil could reduce the crystallization temperature of emulsion.

## 4. Conclusions

The research showed that the interaction between particles and the highly viscoelastic interface film formed were the key factors for determining the freeze–thaw stability of the Pickering emulsion. The higher SPI–maltose concentration made the particles develop a larger size, higher H_0_, and firmer interior integrity. Gel-like flow behavior was exhibited by all the initial emulsions. Emulsions with higher oil phase volume and particle concentration possessed higher viscosity, lower fluidity and a viscoelastic layer. Freeze–thaw stability of these emulsions changed with the cycles of freeze–thaw, particle concentration, and dispersed phase volume fraction. Compared with the emulsion with lower particle concentration, the emulsion with higher particle concentration can prevent emulsion and coalescence, thus showing better freeze–thaw stability. Higher oil or higher particle concentration phase volume has better freeze–thaw stability than that of lower particle concentration and lower oil phase volume. This phenomenon may be generated by Pickering steric stability and the network structure of gel-like. These findings are of great significance for the preparation of high Pickering emulsions with the potential for freeze–thaw stability, which can be used in the production of frozen foods.

## Figures and Tables

**Figure 1 foods-11-04018-f001:**
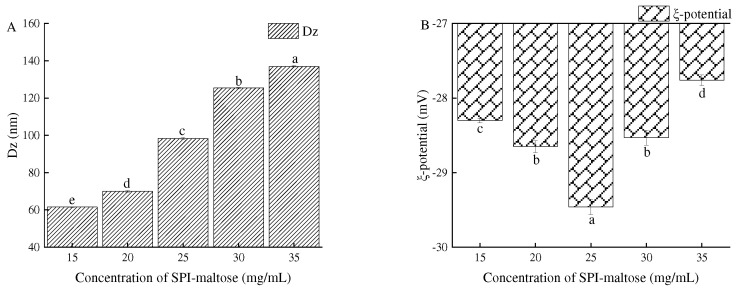
Dz and ξ-potential of different concentration of SPI–maltose. (**A**) shows the particle size, and (**B**) shows the potential. Different letters (a–e) indicate statistically significant differences (*p* ≤ 0.05), according to ANOVA (one-way) and the Tukey test.

**Figure 2 foods-11-04018-f002:**
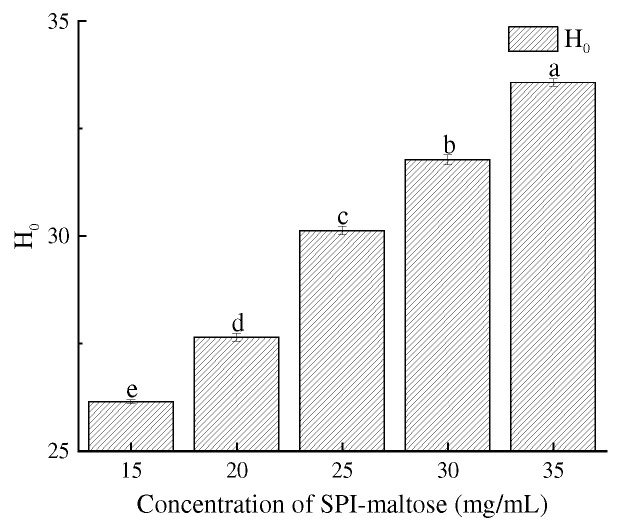
H_0_ at different concentration of SPI–maltose. Different letters (a–e) indicate statistically significant differences (*p* ≤ 0.05), according to ANOVA (one-way) and the Tukey test.

**Figure 3 foods-11-04018-f003:**
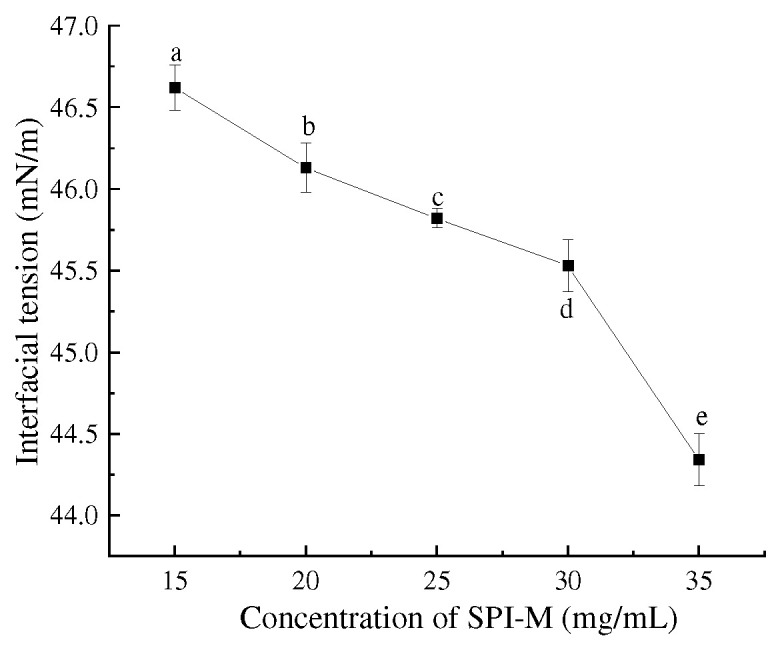
Interfacial tension value of SPI–M prepared dispersion. Different letters (a–e) indicate statistically significant differences (*p* ≤ 0.05), according to ANOVA (one-way) and the Tukey test.

**Figure 4 foods-11-04018-f004:**
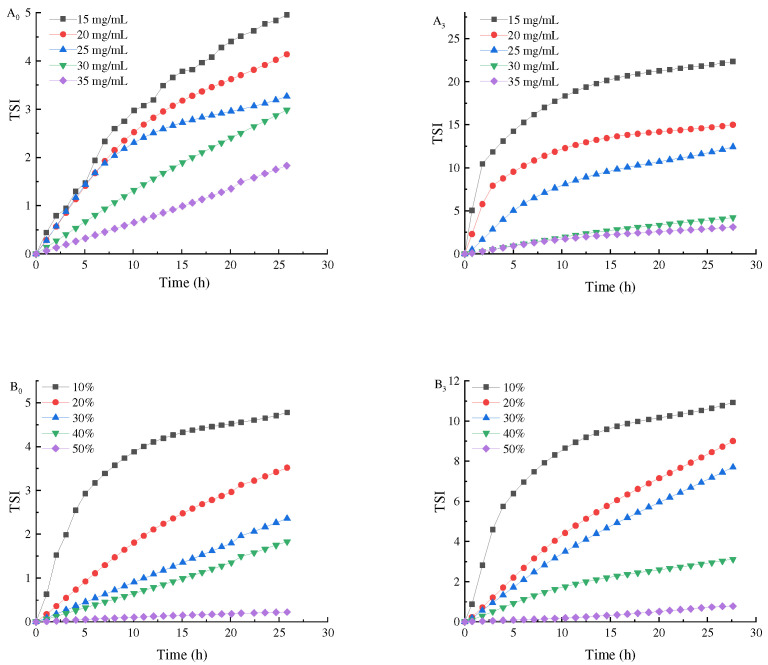
TSI of Pickering emulsions made by different oil phase volume fractions and different concentrations of SPI and maltose. ((**A_0_**) prepared the initial emulsion with different concentrations of SPI and maltose, (**B_0_**) prepared the initial emulsion with different oil phase volume fractions, (**A_3_**) was the emulsion after three freeze-thaw cycles with different concentrations of SPI and maltose, (**B_3_**) was the emulsion after three freeze-thaw cycles with different oil phase volume fractions).

**Figure 5 foods-11-04018-f005:**
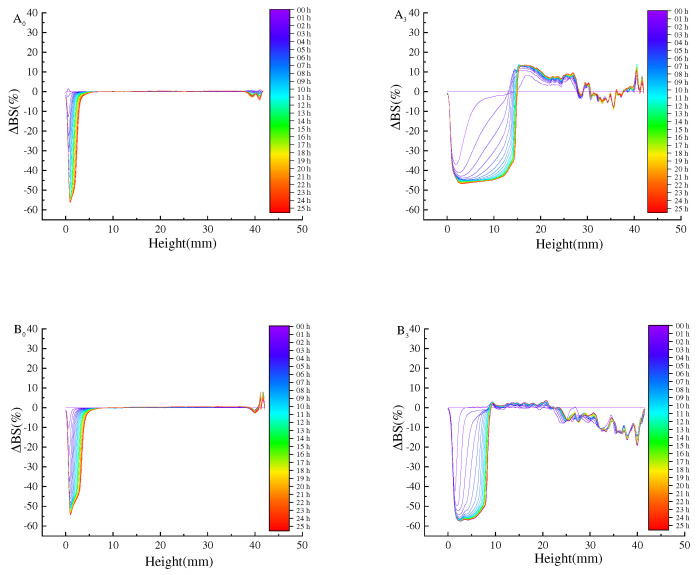
The change of ΔBS under different particle concentrations under the same oil phase volume ratio. (The ΔBS of the Pickering emulsion was measured at 0 h intervals for 25 h across the height of the samples. The blue line was detected at 0 min and the red line was at 25 h. (**A_0_**) 15 mg/mL initial Pickering emulsion, (**B_0_**) 20 mg/mL initial Pickering emulsion, (**C_0_**) 25 mg/mL initial Pickering emulsion, (**D_0_**) 30 mg/mL initial Pickering emulsion, (**E_0_**) 35 mg/mL initial Pickering emulsion, (**A_3_**) 15 mg/mL Pickering emulsion with three freeze-thaw cycles, (**B_3_**) 20 mg/mL Pickering emulsion with three freeze-thaw cycles, (**C_3_**) 25 mg/mL Pickering emulsion with three freeze-thaw cycles, (**D_3_**) 30 mg/mL Pickering emulsion with three freeze-thaw cycles, (**E_3_**) 35mg/mL Pickering emulsion with three freeze-thaw cycles).

**Figure 6 foods-11-04018-f006:**
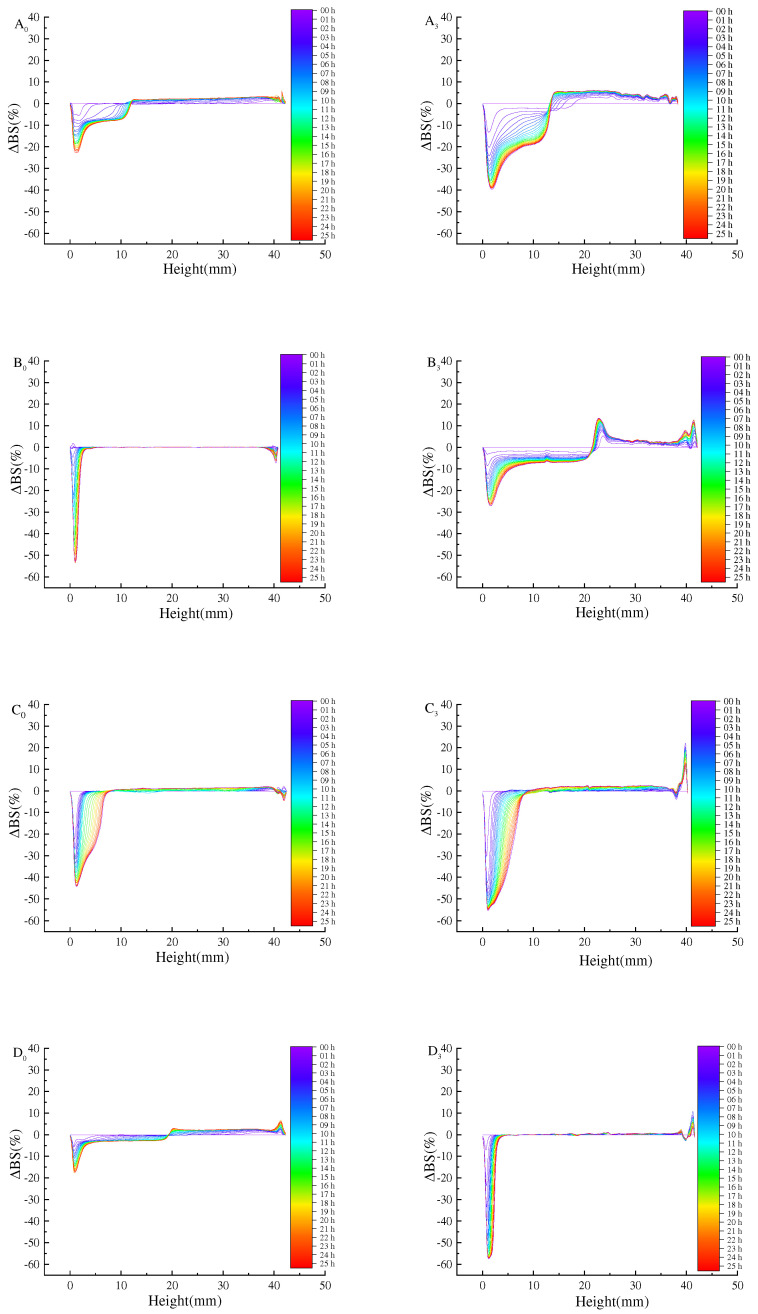
Shows the change of ΔBS of the Pickering emulsion formed by different oil phase volume fractions under the condition of particle concentration 35 mg/mL and after freeze–thaw treatment. (The details are the same as the notes in Figure 5. (**A_0_**) 10% initial Pickering emulsion, (**B_0_**) 20% initial Pickering emulsion, (**C_0_**) 30% initial Pickering emulsion, (**D_0_**) 40% initial Pickering emulsion, (**E_0_**) 50% initial Pickering emulsion, (**A_3_**) 10% Pickering emulsion with three freeze-thaw cycles, (**B_3_**) 20% Pickering emulsion with three freeze-thaw cycles, (**C_3_**) 30% Pickering emulsion with three freeze-thaw cycles, (**D_3_**) 40% Pickering emulsion with three freeze-thaw cycles, (**E_3_**) 50% Pickering emulsion with three freeze-thaw cycles).

**Figure 7 foods-11-04018-f007:**
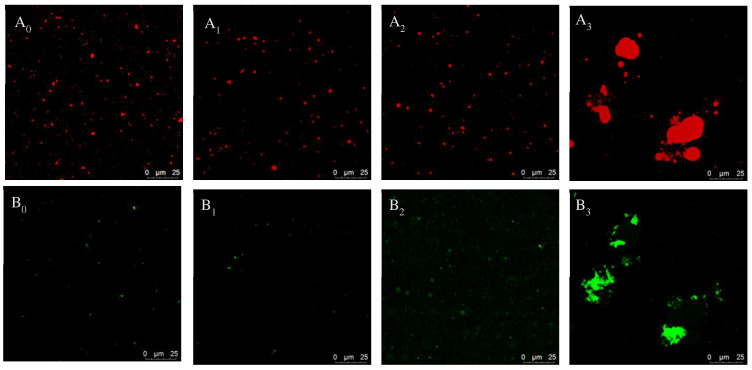
CLSM figures of the Pickering emulsion gels before and after three freeze–thaw cycles. (**A_0_**) initial Pickering emulsion oil phase, (**A_1_**) Pickering emulsion oil phase after one freezing thawing cycle, (**A_2_**) Pickering emulsion oil phase after two freezing thawing cycle, (**A_3_**) Pickering emulsion oil phase after three freezing thawing cycle, (**B_0_**) initial Pickering emulsion protein phase, (**B_1_**) Pickering emulsion protein phase after one freezing thawing cycle, (**B_2_**) Pickering emulsion protein phase after two freezing thawing cycle, (**B_3_**) Pickering emulsion protein phase after three freezing thawing cycle).

**Figure 8 foods-11-04018-f008:**
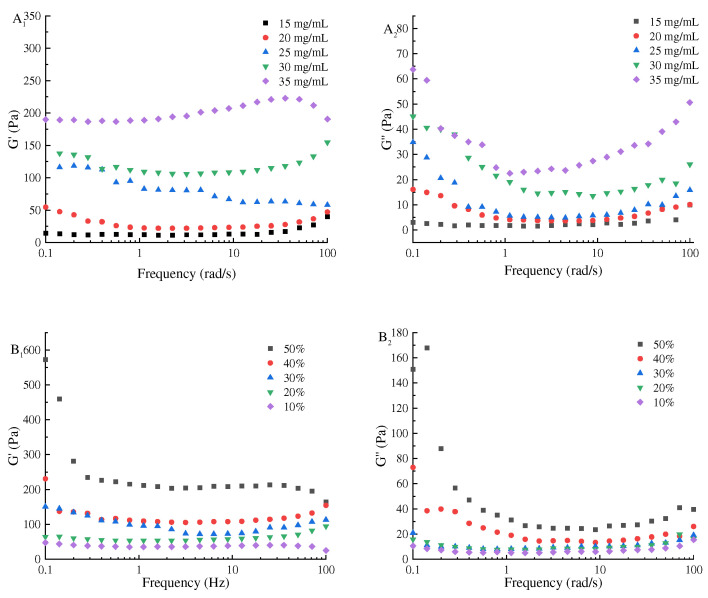
The influence of concentration of SPI and maltose (**A**) and oil phase volume fraction (**B**) on the rheological properties of the Pickering emulsion. (**A_1_**) was the G′ of Pickering emulsion prepared under different concentrations of SPI and maltose, (**B_1_**) was the G′ of Pickering emulsion prepared under different oil phase volume fraction, (**A_2_**) was the G″ of Pickering emulsion prepared under different concentrations of SPI and maltose, (**B_2_**) was the G″ of Pickering emulsion prepared under different oil phase volume fraction).

**Figure 9 foods-11-04018-f009:**
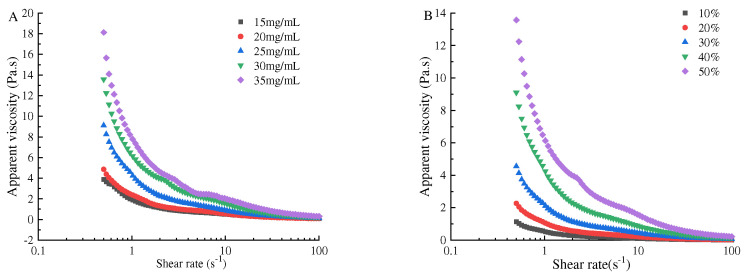
The effect of particle concentration (**A**) and oil phase volume fraction (**B**) on the apparent viscosity properties of the Pickering emulsion. ((**A**) is the apparent viscosity of Pickering emulsion prepared at different concentrations of SPI and maltose, (**B**) is the apparent viscosity of Pickering emulsion prepared at different oil phase volume fraction).

**Figure 10 foods-11-04018-f010:**
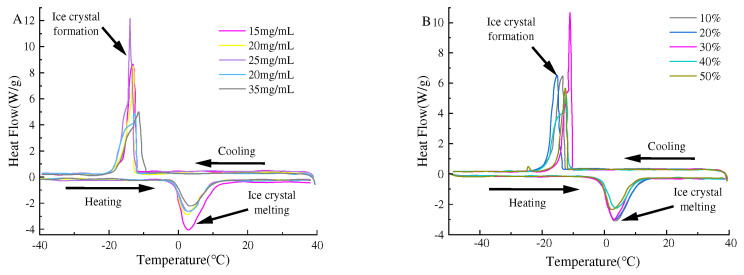
The effect of particle concentration (**A**) and oil phase volume fraction (**B**) on the apparent viscosity of the Pickering emulsion. ((**A**) is the DSC of Pickering emulsion prepared at different concentrations of SPI and maltose, (**B**) is the DSC of Pickering emulsion prepared at different oil phase volume fraction).

**Table 1 foods-11-04018-t001:** Power law equations and coefficients of the Pickering emulsion prepared at different concentrations of SPI and maltose.

Concentrations (mg/mL)	Shear Rate (S^−1^)		
	0.5–100		
	N	K/(Pa·S^n^)	R^2^
15	0.229 ± 0.012 ^a^	2.123 ± 0.137 ^e^	0.982
20	0.225 ± 0.010 ^b^	2.528 ± 0.186 ^d^	0.987
25	0.230 ± 0.011 ^a^	4.537 ± 0.157 ^c^	0.989
30	0.220 ± 0.009 ^c^	6.886 ± 0.264 ^b^	0.984
35	0.191 ± 0.007 ^d^	8.663 ± 0.356 ^a^	0.978

Values are expressed as mean ± expanded uncertainty limit (EUL). ^a–e^ Various superscripts in the same column stand for great diversities (*p* < 0.05).

**Table 2 foods-11-04018-t002:** Power law equations and parameters of the Pickering emulsion prepared at various oil phase volume fraction.

Oil Phase Volume Fraction/(%)	Shear Rate (S^−1^)		
	0.5–100		
	N	K/(Pa·S^n^)	R^2^
10	0.148 ± 0.003 ^b^	0.567 ± 0.023 ^e^	0.989
20	0.148 ± 0.002 ^b^	1.13426 ± 0.054 ^d^	0.989
30	0.148 ± 0.001 ^b^	2.26852 ± 0.126 ^c^	0.989
40	0.148 ± 0.004 ^b^	4.53703 ± 0.275 ^b^	0.989
50	0.230 ± 0.014 ^a^	6.88594 ± 0.327 ^a^	0.984

Values are expressed as mean ± expanded uncertainty limit (EUL). ^a–e^ Various superscripts in the same column stand for great diversities (*p* < 0.05).

## Data Availability

Data is contained within the article.

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
