# Peer review of "Effects of Concentration of Soybean Protein Isolate and Maltose and Oil Phase Volume Fraction on Freeze–Thaw Stability of Pickering Emulsion"

_foods, 2022, doi:10.3390/foods11244018_

Round 1

Reviewer 1 Report

L49: What about the fat crsytallization that can influence the stability?

L 55: compare this system with a simple system containing particles formed via electrostatic interaction, you may use this reference in your introduction: https://doi.org/10.1016/j.foodhyd.2018.10.009

L 64 why not other proteins: for instance flaxseed or pea protein? you may use this references to compare with soy protein: https://doi.org/10.1016/j.fufo.2022.100193

did you consider the water insoluble fraction of soy protein?

can you generalize the surface tension to interfacial tension whereas the lipid phase has different properties than air phase?

Figure 1, it would be better to separate these plots, hard to follow

L189 how would you explain when the concentration is almost doubled but the surface tension increased only few values?

Please add the picture of your emulsions

CLSM images are not clear, IDK if they have any added value to support

Author Response

26-Nov-2022

Dear Reviewers:

Thank you for your letter and the comments on reviewers concerning our manuscript entitled “The effect on freeze-thaw stability of SPI and maltose self-assembled nanoparticles concentration and oil phase ratio” (ID: foods-1998271). Those comments are all valuable and extremely helpful for revising and improving our paper, and they also have an important guiding significance on our researches. We have studied your comments carefully and made corrections and revised portions are highlighted by yellow. We sincerely hope this manuscript will be finally acceptable to be published on the FOODS.

Reviewer 2 Report

Comments to the Author,

I consider the manuscript " Effects of concentration of soybean protein isolate and maltose and oil phase volume fraction on freeze-thaw stability of Pickering emulsion" is interesting. However, as I explain in the corrections and observations made, there are many aspects should be consider in order enhancing the research.

Here are some comments for the authors:

Abbreviations need to be clarified.

Lines 23 – 24. “The results showed that the freeze-thaw stability of the Pickering emulsion prepared with the highest concentration of SPI and maltose increased”

Lines 46 to 49. It is requested to rewrite the idea.

Line 71. It is suggested to correct the spelling mistake.

It is requested to improve the wording of the objective.

Line 155. “The above experiments were carried out at least in triplicate”.

Line 190 onwards. I suggest to improve the explanation about the effect of surface tension on freezing-thawing stability of Pickering emulsion.

In Fig. 3. ¿What does it refer to: A0, A3, B0, B3? 3: ¿is the number of the freeze-thaw cycles? ¿What does it refer to: 10 % to 50%? It should be clarified in the legend of the graph for a better interpretation of it.

Review the legend of Figure 7

Line 328 to 330. “Secondly, the apparent viscosity of emulsion prepared with different concentrations of SPI, maltose and dispersed phase volume fraction η and shear rate γ, the power-law equation was obtained by fitting the relationship between”. The idea is incorrectly written.

Line 338. “This is because Pickering emulsion is composed of solid particles, oil and water”. What is the idea? What do the authors try to explain with that sentence?

Line 339. On the one hand, solid particles are adsorbed at the oil-water interface, and the oil is dispersed in water. On the other hand, SPI maltose molecules have long molecular chains…”. Revise the complete idea.

The DSC results explanation are not clear.

It is suggested the manuscript be reviewed by a native english speaker in order to improve it comprehension.

Some reference are repeated, for example:

5 Zhu, X. F., Zhang, N., Lin, W. F., et al. (2017). Freeze-thaw stability of Pickering emulsions stabilized by soy and whey protein particles. Food Hydrocolloids, 69, 173-184.

8 Zhu, X. F., Zhang, N., Lin, W. F., et al. (2017). Freeze-thaw stability of Pickering emulsions stabilized by soy and whey protein particles. Food Hydrocolloids, 69, 173-184.

Author Response

(The authors gave the same response as above.)

Round 2

Reviewer 1 Report

It is improved